# Novel Micronized Mica Modified Casein–Aluminum Hydroxide as Fire Retardant Coatings for Wood Products

Mezbah Uddin [1], Maitham Alabbad [2], Ling Li [2], Olli Orell [3], Essi Sarlin [3] and Antti Haapala [1,*]

1    School of Forest Sciences, University of Eastern Finland, FIN-80100 Joensuu, Finland; mezbu@uef.fi
2    School of Forest Resources, University of Maine, Orono, ME 04473, USA;
     maitham.alabbad@maine.edu (M.A.); ling.li@maine.edu (L.L.)
3    Department of Materials Science and Environmental Engineering, Tampere University, FIN-33014 Tampere,
     Finland; olli.orell@tuni.fi (O.O.); essi.sarlin@tuni.fi (E.S.)
*    Correspondence: antti.haapala@uef.fi

**Abstract:** Sustainable coating solutions that function as a fire retardant for wood are still a challenging topic for the academic and industrial sectors. In this study, composite coatings of casein protein mixed with mica and aluminum trihydroxide (ATH) were tested as fire retardants for wood and plywood; coating degradation and fire retardancy performance were assessed with a cone calorimeter, and a thermogravimeter was used for the thermal stability measurement. The results indicated that casein–mica composites were beneficial as coatings. The heat release rate (HRR) and the total heat released (THR) of the sample coated with casein–mica composite were reduced by 55% and 37%, respectively; the time to ignition was increased by 27% compared to the untreated sample. However, the TTI of the sample coated with the casein–mica–ATH composite was increased by 156%; the PHR and THR were reduced by 31% and 28%, respectively. This is attributed to the yielded insulating surface layer, active catalytic sites, and the crosslink from mica and endothermic decomposition of ATH and casein producing different fragments which create multiple modes of action, leading to significant roles in suppressing fire spread. The multiple modes of action involved in the prepared composites are presented in detail. Coating wear resistance was investigated using a Taber Abrader, and adhesion interaction between wood and a coated composite were investigated by applying a pull-off test. While the addition of the three filler types to casein caused a decrease in the pull-off adhesion strength by up to 38%, their abrasion resistance was greatly increased by as much as 80%.

**Keywords:** composite coating; flame retardancy; thermal analysis; wear testing; wood

## 1. Introduction

Wood is a traditional building material in the forms of dimension lumber and wood-based composites (e.g., oriented strand board and plywood) which has been dominantly used in light wood-frame residential and commercial buildings for a century. In the past decades, emerging new mass timber products like cross-laminated timber have been on a rapid growth trajectory in mid/high-rise buildings to replace steel and concrete due to unparalleled environmental benefits such as low embodied energy and carbon negativity [1,2]. However, untreated solid wood is rated as a combustible and fire-hazardous material [3].

The design and construction of wood buildings shall meet the requirements of fire safety and fire prevention by complying with a specific fire code; e.g., the 2018 international fire code (IFC) [4]. Effective fire protection measures can provide wood with the ability to withstand flame and its surface propagation, preventing the free access of oxygen which promotes wood destruction and accelerates the burning process [5]. Similar risks have been identified for furniture uses [6] where the chemical treatments of frames and upholstery are now mandatory.

One strategy is to shield the timber structures with non-flammable materials like gypsum boards, leaving the timber only for encased framing. Alternatively, waterborne

fire-retardant (FR) coatings (transparent or opaque) can be applied to wood surfaces because of their porous structure and hygroscopicity. The coating forms a protective layer that can reduce the formation of combustible volatile products, inhibit gas-phase flame reactions, eliminate the surface combustion of carbonaceous residue, and reduce the rate of weight loss [7–9]. When inorganic chemical additives—such as minerals and metal hydroxides—are added to the FRs, a thick coating layer formed may also contribute to slowing down material heating and maintaining the material functions in the fire for a definite period [5,10]. Besides, the ideal coating materials should be easily applied and demonstrate good adhesion to wood and wear resistance.

Halogenated and polyurethane-based fire retardants [11,12] are commercially available, but not all of them are effective or eco-friendly [9]. Halogenated fire retardants have been used extensively in the past and it was noted that under high temperatures, they produce halogen gases that may cause health problems [13–19]. Similarly, polyurethane-based fire retardants release hydrogen cyanide and carbon monoxide, both toxic in terms of health and the environment [20]. The contemporary studies hence focus on novel treatments that combine both efficient fire retardancy and low risk for detrimental emissions. Some of the recent successes include the use of modified lignin [21], natural proteins [22], minerals [23,24], and even modified nanocelluloses [25,26].

Chemical additives such as magnesium hydroxide, aluminum (tri-)hydroxide (ATH), and borate can markedly improve the fire retardancy of wood and different composite materials, including polyurethane-based composites [11,12]. Mineral additives such as vermiculite sodium silicate [27] and montmorillonite [25] in composite coatings have shown good flame retardancy for wood. Recently, a group of mica-based composites has been investigated as fire retardants for different organic polymers and composite materials [28–30]. To the best of the authors' knowledge, little attention has been directed to the application of mica as a functional filler for fire retardant coating formulations intended for wood.

Mica is a dioctahedral silicate mineral comprising two $SiO_4$ tetrahedral sheets and interstratified $AlO_2(OH)_4$ octahedral sheets with the general chemical formula $R^+R_3^{2+}[AlSi_3O_{10}]\cdot(OH)_2$ or $R^+R_3^{3+}[AlSi_3O_{10}]\cdot(OH)_2$ ($R^+ = K^+$, $Na^+$; $R^{2+} = Mg^{2+}$, $Fe^{2+}$, $Mn^{2+}$; $R^{3+} = Al^{3+}$, $Fe^{3+}$, $Mn^{3+}$). One of the few studies in which it has been an active ingredient in an FR solution was by Limparyoon et al. [31], who used poly(AM-co-AMPS-Na$^+$)-mica nanocomposite as a wood coating, obtaining a 31% decrease in the heat release rate. Mica's thermal insulation capacity, hydrophobicity, and capacity to form composites with both inorganic and organic compounds—through chemical and physical interaction [28]—make it attractive for FR formulations. Mica can also block the heat transfer through the surface and promote char formation that acts as an oxygen-capturing agent while also playing a role as an intumescent hydrophobic agent. The aim here is to apply mica and ATH onto wood as a composite, with casein serving as the binding agent for these minerals.

The role of casein protein in FR-coating arises from four different phosphoprotein structures: α-s1, α-s2, β, and κ with clusters of calcium and phosphorus (calcium phosphor-caseinate) forming a very stable colloidal micellar dispersion that has the characteristics of film-formation, crosslinking, and high adhesion [22]. Casein is a phosphoprotein polymer that contains a long carbon chain, phosphate, and amino acids, which comprise two functional groups, e.g., amine and carboxylic acids. During the decomposition of casein, phosphate converts to phosphoric acid, which then catalyzes the dehydration of polymer, promotes char formation [32], converts carboxylic acid to carbon dioxide, which functions as a blowing agent for producing intumescent behavior, and converts the amine to inert ammonia gases which dilute the volatile combustible gases. These actions generated by casein can play significant roles in suppressing fire spread.

We hypothesized that based on the interaction pattern between protein and metal ions, metalloprotein and metallocomplexes are formed as acidic proteins that are especially prone to adsorb to aluminum hydroxides [33] and aluminum ions [34]. The factors associated with a metal ion interaction with proteins are its oxidation state, the radius of the metal ion,

and charge-accepting capacity [35], but the interaction strength of the functional groups and tendency to form coagulation by itself also plays a role [36]. Based on the Lewis acid-base theory, an aluminum ion can form a coordination bond with active sites of ligands, resulting in a distorted octahedral monodentate complex structure [37], an example of which is shown in Figure 1. Other forms of ligand-complexes [37,38] have previously been presented.

**Figure 1.** The possible formation of casein–aluminum ion complex, the so called metalloprotein formation.

Multiple types of physical and chemical interaction between protein and metal ions can take place simultaneously, some of which facilitate the formation of metallocomplexes. One such case is presented in Figure 2. In addition, the adsorption of amino acids in protein to aluminum hydroxide can also take place through ligand exchange mechanisms (Figure 3); e.g., inner-sphere complex formation between aluminum ions and phosphate ions in casein [39–42].

The goal of this study was to develop an FR formulation that is comprised of mica, aluminum trihydroxide, and casein protein with a prior application mostly in natural wood adhesives. The role of casein protein in FR-coating arises from four different phospho-protein structures: $\alpha$-s1, $\alpha$-s2, $\beta$, and $\kappa$ with clusters of calcium and phosphorus (calcium phosphor–caseinate) forming a very stable colloidal micellar dispersion that has the characteristics of film-formation, crosslinking, and high adhesion [22].

**Figure 2.** The possible metallocomplex formation between a metalloprotein and an aluminum ion.

**Figure 3.** The inner sphere complex formation reactions between an aluminum ion and a phosphate group in casein.

## 2. Materials and Methods

Four types of FR composite materials were formulated and coated with the wood substrates. Their thermal stability and fire retardancy were evaluated in terms of thermal gravimetric analysis and cone calorimetric analysis. Also, pull-off strength and abrasion resistance of the prepared composite coatings were assessed.

### 2.1. Materials

Scots pine (*Pinus sylvestris* L.) and spruce (*Picea abies*) were selected as target objects, which are representative softwood species for dimension lumber and wood-based composite panels in building systems. Kiln-dried Scots pine (sapwood) was cut to the dimensions of 100 mm × 100 mm × 20 mm. Three-layer plywood made of spruce was purchased from a local store, which was then processed into samples of the same size. Then, both the solid wood and plywood samples were conditioned in an environmental chamber for one week at room temperature and an RH of 50% to achieve a constant weight.

Sodium hydroxide (NaOH) (≥97% pellet; Merck, Burlington, VT, USA), casein from bovine milk C7078 (technical grade with 0.8–0.9% phosphorus), and aluminum hydroxide $Al(OH)_3$ (reagent grade, 50–57.5%), both from Merck KGaA (Darmstadt, Germany), and mica powder MicaSilkM800 grade (particle size <5 μm) from Dean + Tranter Ltd. (Fordingbridge, UK) were used directly as delivered. Deionized water was used during the experiment when needed.

As an international collaboration, the formulation of FR composite coating materials and thermal performance evaluation were carried out at the University of Eastern Finland and the University of Tampere. The mechanical properties of coatings, pull-off strength, and abrasive wear resistance tests were conducted at the University of Maine, USA to whom the three-layer spruce coated with different FRs were shipped.

### 2.2. Coating Composite Preparation

Suspension and composites of casein were prepared following the method reported by Uddin et al. [43] but with the same ratio of filler-to-casein considered to be the highest additive load that still forms a continuous composite suspension with dissolved casein. To prepare a batch of coating, 2.5 g of casein and 0.1 g of sodium hydroxide (NaOH) were introduced to 12.5 mL of water at 60 °C and stirred by an overhead stirrer on top of a heat plate for 3 min, resulting in a homogeneous suspension. Finally, 4 g of the selected inorganic compound was added to the suspension while stirring for another 3 min to form a composite. Casein–mica, casein–aluminum hydroxide, and casein–mica–aluminum hydroxide composite coatings were prepared using this approach. Once prepared, the wood sample surface was coated with the composites with the help of an adjustable coating blade and dried at room temperature for one week. Coating compositions and the weight

of the coating on the surface of each sample are listed in Table 1. The coated dried wood samples were kept at 22 °C and 50% RH in a fixed humidity chamber for 7 days and then used for thermal tests.

**Table 1.** The composition of composite coatings on a tested wood specimen.

| Sample Code | Coating Composition | Wt of Wood (g) | Wt after Coating (g) | Coating Wt (g) | Coating Wt (g/m$^2$) |
|---|---|---|---|---|---|
| Reference | Uncoated pinewood | 80.50 | - | - | - |
| Casein | 2.5 g C suspension | 99.50 | 100.95 | 1.45 | 145.00 |
| Casein–ATH | 2.5 g C + 4 g ATH | 100.95 | 105.98 | 5.03 | 502.60 |
| Casein–Mica | 2.5 g C + 4 g mica | 110.39 | 116.86 | 6.47 | 647.05 |
| Casein–ATH–Mica | 2.5 g C + 1 g ATH + 3 g mica | 121.18 | 126.57 | 5.39 | 538.66 |

*2.3. Thermal Gravimetric Analysis*

The thermal properties of the samples were studied using a thermogravimetric analyzer (TGA/STDA 851e/LF/1100; Mettler Toledo, Greifensee, Switzerland). Approximately 10 mg of the dry coating were taken from the coated wood samples using a sharp blade to be used for TGA measurement. Thermogravimetric measurement was run under a nitrogen flow rate of 50 mL/min, a heating rate of 10 °C/min, and a temperature range of 25 to 800 °C. The sample (approx. 9–10 mg) in solid form was kept in an open alumina pan (70 μL).

*2.4. Cone Calorimetric Analysis*

Flammability and burning characteristics of the samples were analyzed with a cone calorimeter (Dual Cone Calorimeter, Fire Testing Technology, East Grinstead, UK and ConeTool 1; SGS Govmark Ltd., Farmingdale, IL, USA) at a radiant flux of 50 kW/m$^2$ based on the ISO 5660-2 Standard Test Method but limited to the first 600 s of the experiment using two parallel samples. The heat release rate was measured based on the oxygen consumption and the flow rate in the combustion product stream. The samples were set up horizontally in the sample holder, and only the coated surfaces of the samples were exposed.

*2.5. Pull-Off Adhesion Test*

The pull-off adhesion strength of the coatings was evaluated with a PosiTest by following the ASTM D4541-17 [44]. Six replicates were tested for each coating system. The test was performed by securing a normal (perpendicular) loading fixture (dolly) to the surface of the coating with an adhesive. After the adhesive was cured, a testing apparatus was attached to the loading fixture and aligned to apply normal tension to the test surface. The coating and adhesive were cut down to substrate (wood). The force applied to the loading fixture was then gradually increased and monitored until a plug of material was detached. Then, the peak load was recorded. The 20 mm diameter dollies were glued to the coating surfaces with two components of ResinLab EP11HT epoxy resin. The specimens were conditioned for 24 h at 23 °C and around 60–65% RH. The strength of each sample was calculated by dividing the peak load by the surface area of the dolly.

*2.6. Abrasion Test*

The abrasion resistance performance test method followed the ASTM D4060-19 [45] with four replicates tested for each coating type. The coated panels were conditioned for at least 24 h at 23 °C and 50% RH prior to testing. In a Taber tester, the coated plywood surface was abraded using rotary rubbing action under controlled conditions of pressure and abrasive action of sandpaper, smooth and 180 grit, under 1000 g load. Each test was terminated when the wood surface was exposed but minimum wood fiber was observed on the sandpaper by an experienced operator. The resulting abrasion marks formed a pattern of crossed arcs over an area of approximately 30 cm$^2$.

Abrasion resistance was calculated as loss in weight at a specified number of abrasion cycles, as average loss in weight per thousand cycles of abrasion, or as number of cycles required to remove a unit amount of coating thickness.

The wear index, I, of the test specimen was computed as Equation (1):

$$I = (L \cdot 1000)/N \tag{1}$$

where

L is the weight loss (g),
N is the number of cycles of abrasion recorded.

## 3. Results and Discussion

### 3.1. Thermogravimetric Analysis of Coatings

The thermal stability of the casein suspension and prepared composite coatings was evaluated using thermogravimetric analysis (TG) provided in Figure 4. From the TG curves, it is determined that mica does not decompose at all in this temperature range. This indicates that mica is the most thermally stable compound among tested materials.

Casein macromolecules have several phosphate groups in the micelle structure, and these phosphate groups decompose similar patterns like ammonium polyphosphate (APP). In addition to this, casein retains moisture which starts to evaporate at ca. 100 °C, producing oligopeptides and amino acids containing carboxylic acids which catalyze the pyrolysis process [22]. The thermal decomposition of casein occurs endothermically in four stages. The first stage involves the removal of water molecules in the temperature range of 45.5–176 °C, where no chemical bond cleavage occurs. In the second stage, volatile gases like carbon dioxide ($CO_2$) and ammonia ($NH_3$) are released and remain as polypeptides in the temperature range of 176–248 °C. In the third stage, polypeptide decomposes into a lower molecular weight polypeptide, hydrogen cyanide, carbon monoxide, and primary residue in the temperature range of 248–380 °C. In the fourth stage, primary residue decomposes into methane, water, and ammonia in the temperature range of 380–680 °C [46]. The TG curves of casein and casein suspension show that the thermal degradation pattern is almost the same for both compounds. Both compounds then begin to experience weight loss at about 150 °C, producing phosphoric acid which facilitates the formation of residue that is stable up to about 500 °C.

The pyrolysis of pure ATH is mainly divided into the following two stages. The ATH first loses its moisture over a temperature range of approximately 40–100 °C. ATH begins to decompose endothermically, releasing water and aluminum oxide over a temperature range of approximately 180–392 °C.

In Figure 4 we can see that the degradation temperature of casein–ATH composite is higher than those of ATH, casein suspension, and pure casein due to the formation of a complex compound with ATH. However, the amount of char residue of casein–ATH composite is lower than for pure ATH due to the presence of casein, which itself is less thermally stable than pure ATH. In the case of casein–mica composite, the decomposition temperature and char residue are higher than all the composites and pure materials, except pure mica. These phenomena indicate that the incorporation of mica and ATH improves the thermal stability of casein.

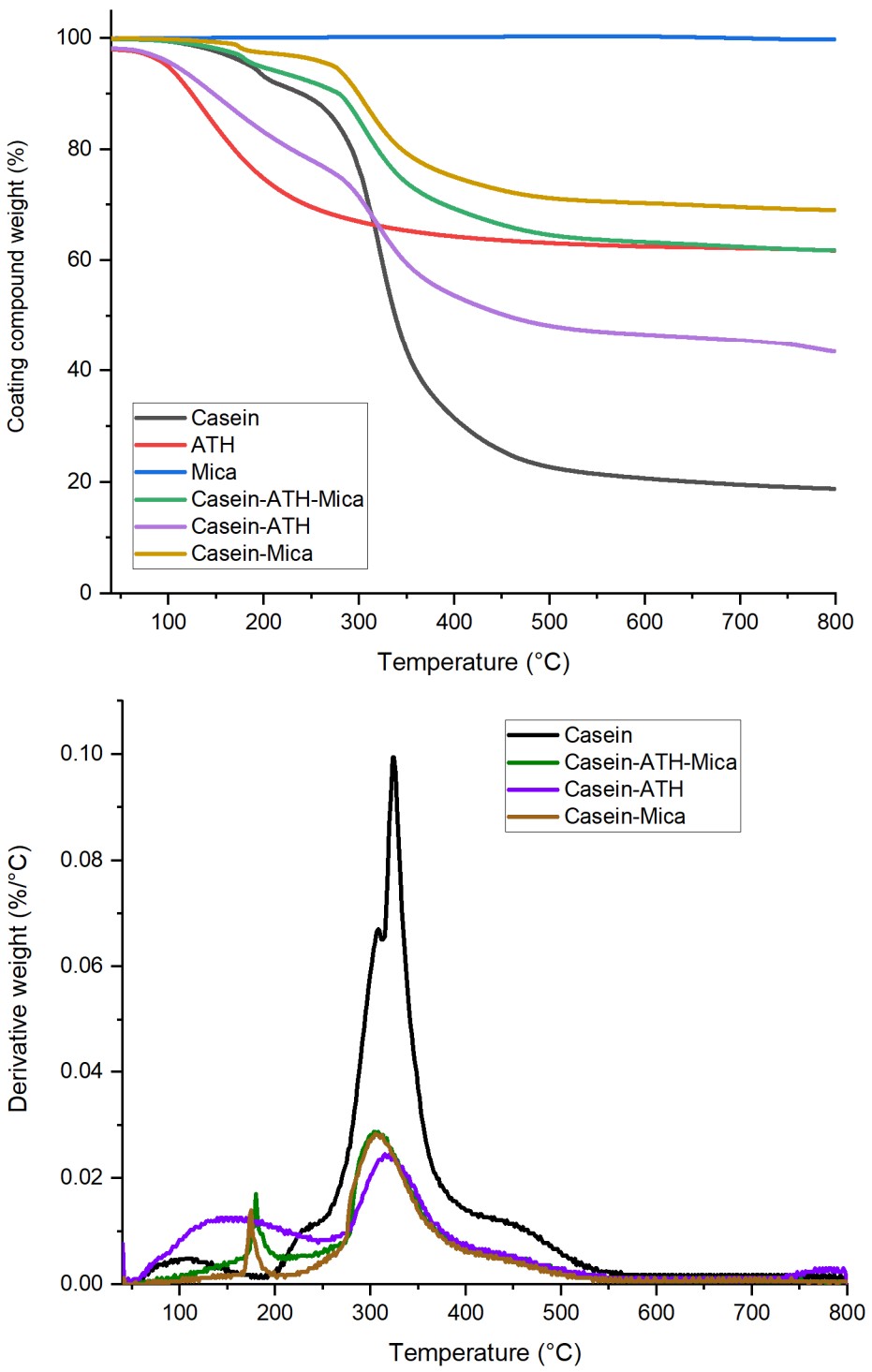

**Figure 4.** TGA (**above**) and DTG data (**below**) from coating compounds as a function of the temperature for pure casein, mica, and ATH, and also mixtures of Casein–ATH–Mica, Casein–ATH and Casein–Mica.

### 3.2. Fire Retardancy Assessment of Coatings

The performance of the prepared composites as wood coatings against fire was assessed by performing a standard cone calorimetry test. The combustion parameters such as time to ignition (TTI), peak heat release rate (PHRR), total heat released (THR), and total mass loss (TML) were evaluated, the results of which are given in Table 2. For some samples, the intumescent behavior of coating was so intense that the swelled coating touched

the sparker, limiting its efficiency. The results indicated that all the coatings (Table 1) can improve fire retardancy of pinewood, except the pure casein suspension, which performed, for most aspects, worse than the untreated reference.

**Table 2.** The calorimetry results (average of two analyses) of an uncoated and composite coated wood specimen with time to ignition (TTI), peak heat release rate (PHRR), total heat released (THR), and total mass loss (TML).

| Sample Code | TTI (s) | PHRR (kW/m$^2$) | THR (MJ/m$^2$) | TML (g) |
|---|---|---|---|---|
| Reference | 12 | 216 | 80 | 47 |
| Casein | 7 | 215 | 86 | 52 |
| Casein–ATH–Mica | 31 | 150 | 58 | 41 |
| Casein–ATH | 42 | 203 | 70 | 46 |
| Casein–Mica | 17 | 97 | 50 | 37 |

3.2.1. Time to Ignition

As can be deduced from the cone calorimetric results, the time to ignition of the samples coated with composites have been prolonged, except wood coated with casein suspension. On the other hand, the TTI of the casein suspension coated sample has been decreased compared to the untreated wood, and a similar trend has been investigated by Alongi et al. (2014) for the cotton fabrics [22]. The reduction of the TTI can possibly result from small inconsistencies in the amount of casein suspension coating on the surface of the wood with respect to other samples. If this were the case, the lower amount of phosphate groups producing a lower amount of phosphoric acid may not be sufficient to actively catalyze the dehydration of hydroxyl groups of wood substrate to promote char formation. The TTI of the casein–ATH composite coated sample is increased from 12 to 42 seconds due to the endothermal decomposition of ATH into aluminum oxide ($Al_2O_3$) and water ($H_2O$), as shown by Equation (2).

$$2Al(OH)_3 \xrightarrow{\Delta} Al_2O_3 + 3H_2O \tag{2}$$

This reaction at the ignition zone is comprised of several processes [47]: The endothermic process makes the material cool, which modifies the pyrolysis process and reduces polymer degradation. The aluminum oxide formed in the ignition process acts as an insulating protective layer on the wood surface, limiting the supply of oxygen and the dissipation of heat. In addition, the water vapor released in the combustion process dilutes the volatile combustible gases, including oxygen. The time to ignition (TTI) of the sample with casein and mica is lower than that with casein and ATH due to the lower specific heat capacity of mica in comparison to ATH. The TTI of the Casein–ATH–Mica mix was also seen to increase with respect to casein–mica due to the incorporation of ATH as a substitute for mica. Although the coating weight was reduced, the combination of these compounds reveals beneficial synergies involved in coating layer performance.

The extension of the TTI of the fire-retardant coatings has been reported previously by several authors. The TTI of the bentonite-infiltrated delignified wood laminate coated with a bentonite nanosheet (34 s) increased by 183% compared to the natural basswood (12 s) when tested at a heat flux of 50 kW/m$^2$ [48]. The extension of the ignition time was attributed to the dense structure of the wood laminate and the heat and oxygen barrier capacity of the bentonite nanosheet. The TTI of the poplar board coated with the coating composite prepared by mixing melamine modified urea–formaldehyde resin, ammonium polyphosphate, and 3A zeolite (279 s) increased by 1760% compared to the untreated sample (15 s) [49]. The TTI of the $TiO_2$/ZnO coated wood (37 s) and ZnO nanorod arrays-coated wood (27 s) increased by 270% and by 170%, respectively, compared to the untreated wood (10 s) [50,51]. However, a comparison with other studies is difficult due to the variable coating thickness, as well as different coating holding substrates and different heat fluxes of the cone calorimeters applied.

### 3.2.2. Peak Heat Release Rate

Heat release rate (HRR) curves of the casein suspension, casein–ATH and casein mica composites are unimodal like untreated wood, as shown in Figure 5. The position of the exothermic peak of the casein suspension and casein–ATH as a composite is yielded ahead of the untreated wood, and the peak value for casein suspension is almost the same as untreated wood. Typically, casein-based FRs can modify the pyrolysis of the wood substrate by generating phosphoric acid, which can catalyze the dehydration of cellulose by phosphorylation, resulting in the formation of a strong char layer and preventing cellulose from decomposing into small molecules. Also, the release of flammable gases, suppressing the further combustion of the substrate, plays a role [32]. The minuscule reduction of peak heat release rate by pure casein coating can be attributed to the formation of cracks on the surface of the coated wood that facilitate rapid heat transfer through the protective coating and into the wood. The low amount of only casein (the lowest concentration of casein with respect to other samples) was not able to catalyze the dehydration effectively for the hydroxyl groups of cellulose and hemicellulose.

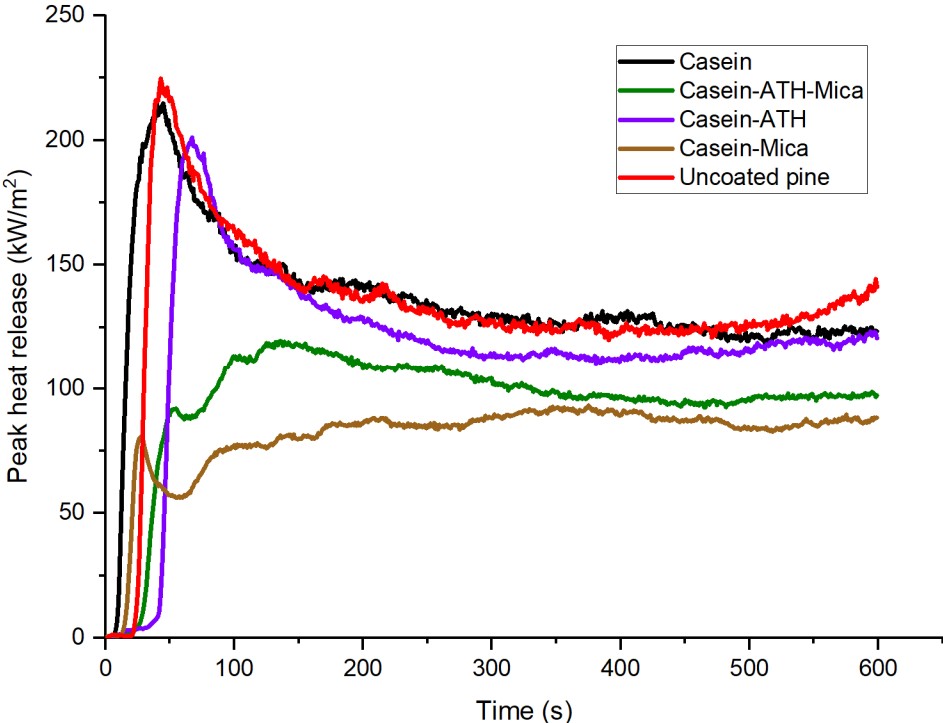

**Figure 5.** The heat release rate for uncoated (reference) and coated wood samples.

Peak HHR, THR, and TML with casein–mica were reduced considerably in comparison to casein–ATH, which can be attributed to the formation of a favorable physical and chemical barrier for heat transfer. Both composites can produce a thermal protective layer on wood from aluminum oxide, but mica can disperse heat more evenly, resulting in an exfoliated nanocomposite with casein [31]. On the other hand, mica slows down the degradation of polymer compounds, including cellulose and hemicellulose, and simultaneously produces a transient aluminosilica protective barrier on the surface of the wood. The combination of these two effects can play a significant role in lowering the PHRR and THR [52]. The minimum amount of total mass loss of casein–mica can be attributed to the mica clay being able to function as a char promoting agent.

The combination of mica and ATH with casein polymer exhibits promising fire retardancy, but for practical application, the performance depends on how well these minerals are dispersed in the polymer matrix. Exfoliated and intercalated nanocomposites could function as a fire retardant more effectively [53,54] than bare chemical mixtures

where micrometer-scale minerals aggregate into bundles; the mechanism of fire protection involves creating multiple degradation pathways to produce more thermally stable compounds. The presence of an exfoliated structure was, however, not investigated here.

Minerals such as mica can also suppress fire spread through physical and chemical actions alone. During burning, mica produces an insulating surface char on the wood laminate, which acts as a thermal and oxygen barrier to lower the HRR. In addition to this, the active catalytic sites of mica can enhance the char formation, as well as catalyze the dehydrogenation and cross-linking of polymer chains [55–57]. These factors play a role in reducing the PHRR. Furthermore, mica can create a complex path for the oxygen diffusive process, leading to an impeding of the escape of volatiles, which then results in the flame resistance of the wood [58,59]. In the mica-polymer nanocomposites, mica acts as a physical crosslink junction within the composite, in which the polymer chain is intercalated within the layers of mica. The intercalating structure of the mica in the nanocomposite hence provides a better barrier against external heat [60].

The chemical actions of fire protection are cleavage polymer, resulting in char occurring in the condensed phase, and cellulose and hemicellulose catalytic dehydration performance by the solid acidity of zeolites like phosphoric acid [61,62]. The textural properties of porous aluminosilicates can play a role during the dehydration of cellulose.

### 3.3. Pull-Off Strength of Coating Systems

The results for the pull-off strength are shown in Figure 6 as box plots defined by the 25th and 75th percentiles. The casein coating system resulted in the highest adhesion of the coating to the plywood surface with a mean of 2.63 MPa, whereas the coating systems with the combinations of casein and mica and casein and ATH, as well as casein, mica, and ATH resulted in lower adhesion of the coating to the plywood surface (1.88 MPa, 1.97 MPa, and 1.79 MPa, respectively). Higher fire retardance performance seems to be inversely proportional to the adhesion strength of the coating, which would imply that with better adhesion the overall durability of these systems could be markedly improved.

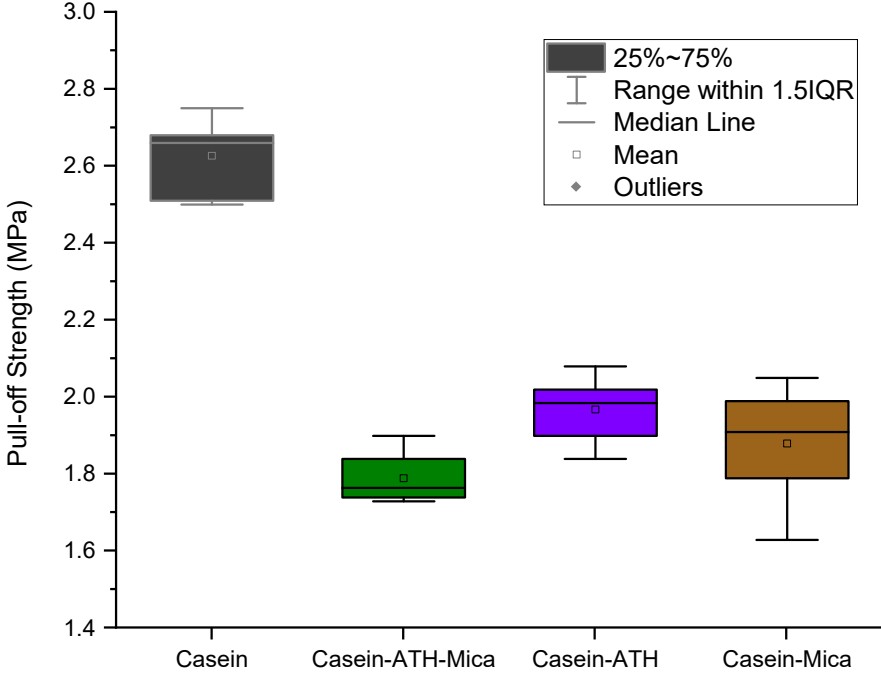

**Figure 6.** The pull-off strength results for the tested coating systems.

Four types of failure modes that can be considered for the pull-off adhesion tests include (a) failure in substrate, (b) bond failure at wood and overlay interface, (c) failure in overlay, and (d) bond failure at epoxy/overlay interface. Figure 7 shows one sample

of the tested specimens for the pull-off test of the coated plywood. We can observe that the tested specimens showed a type (a) failure mode, i.e., failure in the wood substrate, whereas type (c) failure mode—i.e., failure in the coating layer—can be seen in casein and mica, and casein, mica, and ATH coating systems. Casein with ATH was of type (b) failure mode, i.e., bond failure at the wood and coating layer interface. No failure occurred at the interface of the epoxy and coating layer, indicating that all the testing results are reliable. One-way ANOVA analysis was performed. The *p*-value in Table 3 was much lower than 0.05, indicating that the means of four groups have a statistical difference.

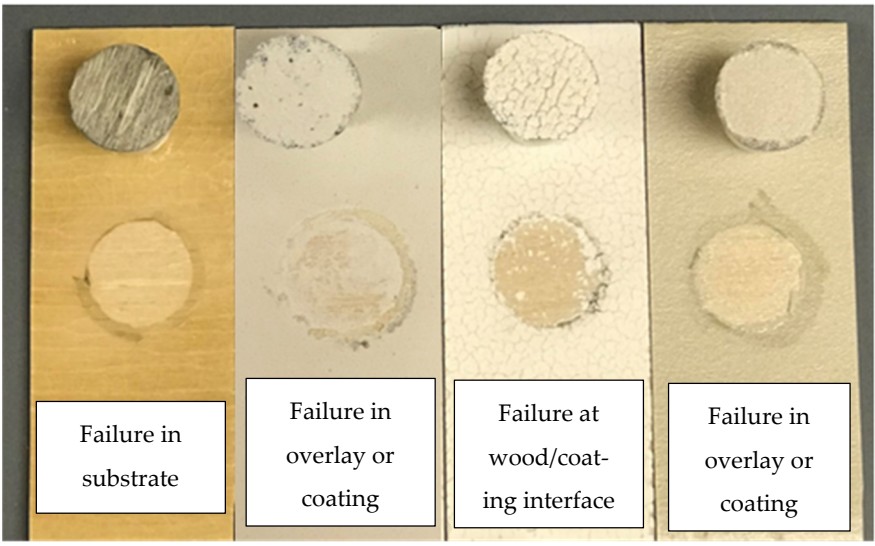

**Figure 7.** Visual summary of pull-off adhesion tests with a coated plywood specimen, from left: casein; casein–ATH–mica, casein–ATH and casein–mica.

**Table 3.** One-way ANOVA analysis of pull-off test results.

| Source of Variation | SS | df | MS | F | *p*-Value | F Crit |
|---|---|---|---|---|---|---|
| Between Groups | 2.607945833 | 3 | 0.869315278 | 77.82009201 | $3.3362 \times 10^{-11}$ | 3.098391212 |
| Within Groups | 0.223416667 | 20 | 0.011170833 | | | |
| Total | 2.8313625 | 23 | | | | |

### 3.4. Abrasion Resistance Performance of Coating Systems

The Taber apparatus was used to measure the abrasion resistance of the coating systems. The average weight loss and the average wear indices of the samples are summarized in Table 4. A high Taber Wear Index indicates a low abrasion resistance, which is also reflected in a high weight loss. The best results are delivered by the casein coating system because of the lowest average wear index of 2.333 g/cycles. The casein and ATH coating system shows a comparable wear resistance performance with the casein and mica coating system. The casein, ATH, and mica composite coating system has the highest index of 4.957 g/cycles. One-way ANOVA analysis was also performed. The *p*-value in Table 5 was much lower than 0.05, indicating that the means of four groups have a statistical difference. Figure 8 shows one sample of the abraded coated plywood surfaces for the eight types of coated plywood.

**Table 4.** A summary of the abrasion resistance results of the coated plywood.

| Sample | Average Weight Loss (g) | Average No. of Cycles | Average Wear Index (g/Cycles) |
|---|---|---|---|
| Casein | 0.420 | 180 | 2.333 |
| Casein–ATH–Mica | 2.018 | 407 | 4.957 |
| Casein–ATH | 1.650 | 3 | 3.811 |
| Casein–Mica | 1.970 | 528 | 3.731 |

**Table 5.** A one-way ANOVA analysis of the abrasion resistance results.

| Source of Variation | SS | df | MS | F | *p*-Value | F Crit |
|---|---|---|---|---|---|---|
| Between Groups | 14.20487 | 3 | 4.734958 | 89.43616 | $1.77 \times 10^{-8}$ | 3.490295 |
| Within Groups | 0.635308 | 12 | 0.052942 | | | |
| Total | 14.84018 | 15 | | | | |

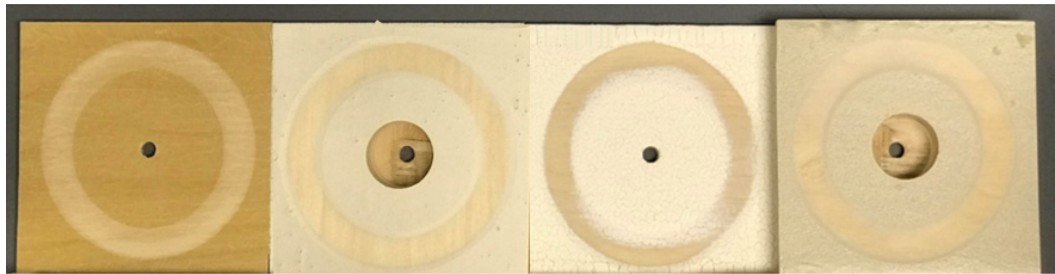

**Figure 8.** Images of the tested plywood specimens using the Taber Abrader, from left: casein; casein–ATH–mica, casein–ATH and casein–mica.

The surface roughness of the four samples visible in Figures 7 and 8 differs from each other due to the moisture absorption capacity, the concentration of the chemicals in the coating composites, the particle size, and interaction capacity with the used chemicals. The coating surface of the casein–mica specimen had no visible cracks due to the coating composition, including mica, which has lower hygroscopicity and smaller particle size, in addition to the large surface area, leading to strong interaction with casein. The surface of the casein–ATH specimen had cracks attributed to the presence of ATH, which is more hygroscopic and absorbs more water, resulting in the coating initially swelling, then shrinking while drying and cracking. However, the incorporation of ATH with mica avoids the formation of surface cracks on the coating surface combining these constituents, which is assumed to be beneficial to the fire retardancy of the coating composite.

## 4. Conclusions

This study shows that the addition of either mica, ATH, or both to casein improved the fire retardancy of the coating composites. The formulated composites functioned as fire retardants for wood by creating multiple modes of physical and chemical actions. For the combination of casein with mica, the resulting composite reduced PHHR by 55% compared to uncoated pinewood. For the combination of casein with ATH, the resulting coating increased TTI by 156% with respect to the untreated pinewood. Comparatively, for the combination of casein with mica and ATH, the resulting composite could decrease PHHR by 31% and increase TTI by 256%. Thermally stable metallic oxide and aluminosilicate barriers resulting from the degradation of coating systems led to the inhibition of the heat and mass transfer between the surface of the wood and the melting coatings, causing a higher fire-retardant performance. The addition of the three filler types to casein caused a decrease in the pull-off adhesion strength by 29–38%; however, their abrasion resistance was greatly increased by 46–72% compared to the reference casein coating layer. The coating composite

with the casein–ATH–mica system could, without the unwanted crack formation, release water vapor and generate the synergistic effect in the casein–ATH composite coating system, which is mainly attributed to physical effects in the condensed phase. Hence, we see a marked improvement in coating systems by introducing components that via different modes of action can enable fire retardance properties to wooden material. From this study, it is seen that even though pure casein suspension (concentration used in this study) showed no improvement, mineral composites can markedly improve the fire retardancy of the wood. However, further research can be conducted to identify potential issues with sustainability challenges related to the use of non-renewables, and also to assess the smoke production volume and health risks associated with their emissions.

**Author Contributions:** Conceptualization, A.H. and M.U.; methodology, M.U., M.A., L.L., O.O., E.S. and A.H.; validation, M.U. and M.A.; resources, L.L., E.S. and A.H.; data curation, L.L., O.O., E.S. and A.H.; writing—original draft preparation, M.U.; writing—review and editing, all authors; visualization, M.A. and A.H.; supervision, L.L. and A.H.; project administration, L.L. and A.H.; funding acquisition, L.L. and A.H. All authors have read and agreed to the published version of the manuscript.

**Funding:** For funding, the authors gratefully acknowledge the support to work of pull-off adhesion and wear resistance tests of coating materials conducted at the University of Maine (USA) by the U.S. Department of Agriculture's (USDA) National Institute of Food and Agriculture (NIFA), McIntire-Stennis Project Number ME042205, through the Maine Agricultural and Forest Experiment Station. The material development work at University of Eastern Finland (Finland) was supported by Academy of Finland (project 329884) and Heikki Väänänen fund.

**Informed Consent Statement:** Not applicable.

**Data Availability Statement:** Not applicable.

**Acknowledgments:** We would like to thank Jesse Savolainen and Kalle Kiviranta for their help with the cone calorimetry tests and Tommi Kokkonen for TGA analyses.

**Conflicts of Interest:** The authors declare no conflict of interest.

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
