# Peer review of "Novel Micronized Mica Modified Casein–Aluminum Hydroxide as Fire Retardant Coatings for Wood Products"

_coatings, doi:10.3390/coatings12050673_

Round 1
Reviewer 1 Report
The work deals with the problem of fire resistance of wood and wood products which is a problem stated by many wood users around the world. Even there are current approaches to deal with the problem, still the practical approach is to increase the size of the wood boards to increase the time that the whole building can stand a fire, but there is a lack of work on increasing the fire resistance of wood as a separate material. This work comes to partially fill that gap offering an option on the finishing that can be applied to the wood to slow down the action of fire.
The paper is well written, provide well detailed information and it is an original peace of work. I only have few comments for the authors.
I would have been relevant to quantify the density of the wood samples, as a high density normally implies more fire resistance. In the manuscript the weight of the samples is given, if we consider the same volume on all samples, density is indirectly provided. However, there is no discussion on how the differences of weight would have affected the results.
In table 1 the proportion of ATH and Mica are the same for the samples treated with Casein-ATH and Casein-Mica respectively. Later, on the samples treated with Casein-ATH-Mica the proportion of ATH/Mica was 1:3. Why did you use such proportion. Why not keeping it 1:1? Was there any stoichiometric reason? If this the case, I would recommend adding it to the manuscript.
The evaluation of the coating including and abrasion test. Why is this property important to be evaluated? Why not evaluate resistance to lixiviation and compatibility with other finishes that would act later as top coat The pull off test seems more than enough to me to check on the compatibility of the coating with the substrate.
The study lacks a statistical analysis beyond that shown for the pull off test. It is relevant to indicate the number of samples tested on the fire test and show at least an average and standard deviation values as results.
Prospect for further research must be included in the discussion or conclusion, similarly state what are the limitation of the technology that is proposed. Surely, the limitations can be addressed in a future work.
Minor changes on the manuscript:
Line 56: “..extensively before and it was noted…”
Lines 127 and 129: be consistent with the units (mm or mm2?)
Line 254: “…but mica can disperse heat more evenly resulting in an exfoliated nanocomposite with casein.”, you to add a reference for such statement.
Line 290: “ca.” what is the meaning of that?
Line 300: “Figure 6” I guess you mean Figure 5.
Other than that, the manuscript is a fine piece of work that suitable to be published in the journal.
Author Response
We have addressed comments in the attached file

Reviewer 2 Report
The review report, manuscript "coatings-1651828", title "Novel Micronized Mica Modified Casein-Aluminum Hydroxide as Fire Retardant Coatings for Wood".
The topic of the manuscript is interesting, the aim of the research is clear. I have a few suggestions for improving the quality of the manuscript before publication:
Please mention also plywood in the Title, as it was part of the research.
Line 18: In case of time to ignition you added results in %, in the case of HRR you talk about significant differences, please discuss all the results the same way.
In the abstract please add the novelty of your research.
Line 30: It should be oriented strand board (not orientated)
Please add some references to your statements in the first paragraph of the introduction.
Please move the goal of the manuscript to the end of the Introduction part.
Also the part in lines 115-121belong to the Materials and Methods part.
LInes 124-131, please discuss the surface properties of the wood and plywood samples.
In the Materials and Methods, the part should be added statistical evaluation of data.
Part 3 should be named Results, then 3.1 Fire Retardancy...
The results part is well written. Please add proper discussion with other authors. Also, please add a discussion about the statistical significance of the results.
In the Conclusions part please discuss the limitations of your research and the novelty of your research.
Author Response
We have addressed comments in the separate file

Reviewer 3 Report
Comments to Authors:
Ref. No.: coatings-1651828
Title: Novel Micronized Mica Modified Casein-Aluminum Hydroxide as Fire Retardant Coatings for Wood
Overview and general recommendation:
In this manuscript, the authors prepared some composite coatings to improve the flame retardancy of wood and plywood. For these composite coatings, they are made of casein protein mixed with mica and/or aluminum trihydroxide (ATH). Then, their reaction-to-fire properties were evaluated using cone calorimeter and their thermal stability were investigated using TGA. After this, pull-off strength and abrasion resistance of the prepared composite coatings were also assessed. This study will give us some insight into the development of non-halogenated based flame retardant coating onto wood to improve its fire safety. However, the authors should further polish some content presented in this manuscript. This manuscript also has some shortcomings in regards to some data analyses.
Major comments:
- Both mica and ATH are commonly used flame retardant additives. In the introduction section, the authors may need to introduce more clearly the novelty of this study, such as why to use casein protein as a novel flame retardant additive here.
- Two and half pages are a little too long for the introduction section. The authors may need to condense the information a little bit.
- For the wight of wood shown in Table 1, why are their values so different? They have the same dimensions, so that they should have the same weight. What causes the difference here?
- For the formula for Casein-ATH-Mica shown in Table 1, how is it determined? Is this the optimal formula for the mixture of Casein, ATH, and Mica?
- For the values shown in Table 2, check the significant digits. For example, for time to ignition, it is common to say 12 s, rather than 12.10 s. “For best samples the time to ignition of casein- mica-ATH composite coating increased by 156.2%” is not a common expression.
- For the curves shown in Figure 4, besides color, it is better to include label for different curves. The same comment can be applied for Figure 5.
- In Line 54, the authors mentioned “polyurethane-based fire retardants”. What do you mean this?
- It might be better to present the TGA results before the sections of 3.1 and 3.2. In this case, based on TGA results, the authors can better discuss the results of cone calorimeter tests.
Author Response

(The authors gave the same response as above.)

Reviewer 4 Report
Fire retardant coatings are very important for wood protection. This study shows that the addition of either mica or ATH or both to casein improved the fire retardancy of the coating composites. The fire retardant performance is successfully and this paper can be publish to Coating journal. There are a few minor issues that need to be noticed before publication.
- Part 3 should be RESULTS AND DISCUSSION. Conclusions should be part 4.
-
Abbreviations need to be modified. There is no need for repeated abbreviations, but first appearance.
- Line 127, may be 100 mm x 100 mm x 20 mm.
- Line 282, I suggest to put TG and DTG in one figure for better comparison of experimental results.
- Figure 8 is not very clear. Can replace the high-resolution pictures?
-
Whether the author has done a real fire test, how about the effect?
Author Response

(The authors gave the same response as above.)

Reviewer 5 Report
Your argumentation line is quite strange, on the one hand you complain about toxic gases which are released by PU coating-based FR and on the other hand you use a protein, which also releases toxic gases by thermal composition? Honestly?
Another big problem, the authors show in their paper that they haven’t understood FR-chemistry. There are essential problems in the whole discussion of the paper, casein is a good FR due to its P content. But the essential work of P in FR is totally not discussed, therefore the carboxylic acid should to the work of P?
4 experiment and no variation in the ratios, just blind test, what kind of science should that be?
Fig. 1. Carboxylic acids are bis-dentade ligands. In Fig 1 they are shown as mono-dentate complex. Is that different in Carboxylic A-L complexes?
If I look to page 6 and the decomposition claim of Mica in eq. 2 and look after wards to the TGA of Mica, whereby on the temperature range no decomposition is happening, I disbelieve the claimed action in this paragraph.
Further strange things:
ATH decomposition temperature following to the thermogram is 150 °C!!!
Something that's is quite strange, ATH decomposition temperature is 220 °C !!!
“In addition to this, casein retains moisture which starts to evaporate at ca. 100 ℃ , producing oligopeptides and amino acids containing carboxylic acids which catalyse the pyrolysis process.”
This section show that the authors did not have a look to the literature! The thermal decomposition is described here:
https://www.sciencedirect.com/science/article/abs/pii/S0040603112003759
And if the author would know more about FR chemistry, then it should be well known, that carboxylic acid easily decompose to CO2, which is normally a blow agent for intumescent coatings.
How much add on in on the wood?
Tabber abrader: is the weight lost just the coating or also wood?
Author Response

(The authors gave the same response as above.)

Round 2
Reviewer 2 Report
The manuscript was improved, I suggest accepting it.
Author Response
We have corrected a few identified issues with the language as no other revisions were required from Rev. #2
Reviewer 5 Report
Reject, no significant improvment.
Failure are still inside.

Author Response
Comments have been addressed in a separate file
